# PGE_2_ Is Crucial for the Generation of FAST Whole- Tumor-Antigens Loaded Dendritic Cells Suitable for Immunotherapy in Glioblastoma

**DOI:** 10.3390/pharmaceutics12030215

**Published:** 2020-03-02

**Authors:** Sara Nava, Daniela Lisini, Simona Frigerio, Simona Pogliani, Serena Pellegatta, Laura Gatti, Gaetano Finocchiaro, Anna Bersano, Eugenio Agostino Parati

**Affiliations:** 1Cell Therapy Production Unit—UPTC and Cerebrovascular Unit, Fondazione IRCCS Istituto Neurologico Carlo Besta, 20133 Milan, Italy; daniela.lisini@istituto-besta.it (D.L.); simona.frigerio@istituto-besta.it (S.F.); simona.pogliani@istituto-besta.it (S.P.); laura.gatti@istituto-besta.it (L.G.); anna.bersano@istituto-besta.it (A.B.); eugenio.parati@istituto-besta.it (E.A.P.); 2Unit of Molecular Neuro-Oncology, Fondazione IRCCS Istituto Neurologico Carlo Besta, 20133 Milan, Italy; serena.pellegatta@istituto-besta.it (S.P.); gaetano.finocchiaro@istituto-besta.it (G.F.); 3Laboratory of Brain Tumor Immunotherapy, Fondazione IRCCS Istituto Neurologico Carlo Besta, 20133 Milan, Italy

**Keywords:** dendritic cells, Fast protocol, immunotherapy, PGE_2_, GBM

## Abstract

Dendritic cells (DC) are the most potent antigen-presenting cells, strongly inducers of T cell-mediated immune responses and, as such, broadly used as vaccine adjuvant in experimental clinical settings. DC are widely generated from human monocytes following in vitro protocols which require 5–7 days of differentiation with GM-CSF and IL-4 followed by 2–3 days of activation/maturation. In attempts to shorten the vaccine’s production, Fast-DC protocols have been developed. Here we reported a Fast-DC method in compliance with good manufacturing practices for the production of autologous mature dendritic cells loaded with antigens derived from whole tumor lysate, suitable for the immunotherapy in glioblastoma patients. The feasibility of generating Fast-DC pulsed with whole tumor lysate was assessed using a series of small-scale cultures performed in parallel with clinical grade large scale standard method preparations. Our results demonstrate that this Fast protocol is effective only in the presence of PGE_2_ in the maturation cocktail to guarantee that Fast-DC cells exhibit a mature phenotype and fulfill all requirements for in vivo use in immunotherapy approaches. Fast-DC generated following this protocol were equally potent to standard DC in inducing Ag-specific T cell proliferation in vitro. Generation of Fast-DC not only reduces labor, cost, and time required for in vitro clinical grade DC development, but can also minimizes inter-preparations variability and the risk of contamination.

## 1. Introduction

Dendritic cells (DC) can be found in different tissues where they form the link between the innate and adaptive immune system. DC are the most potent antigen-presenting cells (APCs) and, as such, are strong inducers of T cell-mediated immune responses. DC detect homeostatic imbalances, process antigens for presentation to T cells [1] and secrete cytokines and growth factors [2]. DC are found in two different functional states: “Mature” and “immature”. These states are distinguished by many features, but the ability to activate antigen-specific naïve T cells in secondary lymphoid organs is the hallmark of mature DC [3].

In their resting state, DC are considered to be immature but primed to acquire antigens. Upon exposure to activating stimuli, DC undergo a series of phenotypic and functional changes referred as “activation” and “maturation” [4]. The process of DC activation is a controlled differentiation process that is closely associated with antigen acquisition; it is characterized by the upregulation of chemokine receptors, adhesion molecules, and costimulatory molecules. DC maturation is characterized by the reduction in phagocytic capacity, enhancement in antigen processing and presentation, improved migration to lymphoid tissues and increase in the capacity to stimulate B and T cells [5]. Maturation is induced by microbial products or by the action of inflammatory molecules (TNFα, IL-1, IL-6, and IFNα) [6].

The role of the immune system in eliminating tumors has been established in several studies. The DC’s central role in initiating immune responses led to the development of immunotherapeutic strategies exploiting the ability of these potent APCs to drive T cells, B cells and natural killer (NK) cells to become antitumor effectors, enhancing their recruitment into the tumor [7]. DC-based immunotherapies are currently being tested against various forms of cancer in clinical settings [8]. With regard to glioblastoma, a large variety of clinical trials employing DC have been published [9], showing increased overall survival [10]. However, DC production for clinical applications is decidedly labor intensive and expensive.

Most clinical protocols for the preparation of DC started from the isolation of circulating monocytes [11]. Fully mature DC are generally obtained in 7–10 days of culture: differentiation step take place in 5–7 days with GM-CSF and IL4; immature DC are subsequently induced to maturation for 1–3 days with pro-inflammatory stimuli to generate a population of immunogenic mature DC [12]. Several groups have shown that it is possible to obtain mature DC in 2–3 days cell culture [13,14,15,16,17]. The in vitro generation of DC with Fast protocol will shorten the time from the patients’ recruiting to DC treatment and therefore be beneficial in the clinical setting: a Fast protocol would result in a simplified processing that minimizes inter-preparations variability and the risk of contamination, also resulting less expensive. Moreover, a Fast protocol may more closely resemble the in vivo development of DC from monocytes [17,18]. The success of immunotherapy approaches based on DC strongly depends on the ability to generate functional mature DC (mDC) with a high level of standardization. In the present study we aimed at developing a Fast-DC method in compliance with good manufacturing practices (GMP) for the production of autologous mDC loaded with antigens derived from whole tumor lysate. Here we report the results from a comparison of our standardized GMP protocol (7 days) and a shorter (3 days) protocol.

## 2. Materials and Methods

### 2.1. Generation of DC from Peripheral Blood Mononuclear Cells (Classical Method—7 Days)

The study was approved by the local institutional review board of the Fondazione IRCCS Istituto Neurologico Carlo Besta (Milan, Italy). Informed written consent was obtained from all patients before procedure.

Standard method DC have been prepared as previously reported [19].

Briefly, patients underwent leukapheresis procedure without cytokine stimulation. The leukapheresis product was washed with PBS/EDTA (Miltenyi, Bergish Gladbach, Germany) and centrifuged at 200 g for 10 min. Total whole blood cells were incubated with anti-CD14-conjugated beads (Miltenyi) for 15 min at room temperature; excess of antibody is washed out prior to sorting on CliniMACS system (Miltenyi). Positive CD14+ fraction was cultured at 3–5 × 10^6^ cells/mL in VueLife closed culture systems (Afc) in CellGRO serum free Medium (CellGenix, Freiburg, Germany), implemented with 20 ng/mL IL-4 and 50 ng/mL GM-CSF (CellGenix).

On day 5 of culture immature DC were pulsed for 24 h with autologous tumor lysate, at the concentration of 30–50 μg/10^6^ living cells, plus 50 μg/mL keyhole limpet hemocyanin (KLH, Calbiochem, San Diego, CA, USA) with further addition of 10 ng/mL IL-4 and 25 ng/mL GM-CSF. On day 6, antigen-loaded DC were cultured with pro-inflammatory cytokines cocktail: 10 ng/mL of TNF-α, IL-1β, IL-6 (CellGenix) with or without addition of PGE_2_ (1 µg/mL, Sigma Aldrich, Darmstadt, Germany). After a further 24 h, mature antigen-loaded DC were collected and frozen. All reagents and disposables were clinical grade certified.

The standard protocol in the presence of PGE_2_ will in the following be referred to as mDCp; the standard protocol in the absence of PGE_2_ will in the following be referred to as mDC.

### 2.2. Generation of DC from Peripheral Blood Mononuclear Cells (FAST Method_3 Days)

Methodological details of CD14+ monocytes isolation and culture (day 1) were done as reported in the Section 2.1. On day 2 of culture, immature Fast-DC were pulsed with autologous tumor lysate as reported in Section 2.1. On day 3, antigen-loaded Fast-DC were induced to maturation with pro-inflammatory cytokines cocktail: 10 ng/mL of TNF-α, IL-1β, and IL-6 (CellGenix), with or without addition of PGE_2_ (1 µg/mL, Sigma Aldrich). After a further 24 h, mature antigen-loaded Fast-DC were collected and frozen. 

The Fast protocol in the presence of PGE_2_ will in the following be referred to as mDCp-F; the Fast protocol in the absence of PGE_2_ will in the following be referred to as mDC-F.

### 2.3. Flow Cytometry

For analysis of DC, the following monoclonal antibodies were used: anti-CD14, anti-HLA-DR, anti-CD80, anti-CD86, anti-CD83, and isotype controls (all from Miltenyi). 1 × 10^5^ cells/tube were stained with fluorochrome conjugated mAbs (BD) and incubated for 30 min at 4 °C in the dark. All labeled cells were analyzed on a FACS Calibur (BD). Data analyses were conducted using CellQuest software (Becton Dickinson Company, Franklin Lakes, NJ, USA).

### 2.4. IL-12(p70) Measurements

Supernatants were collected from all cell culture conditions, at the day of harvest, for IL-12(p70) enzyme linked immunosorbent assay (ELISA) kit purchased from RnD systems and manufacturer’s instructions were followed.

### 2.5. Endocytic Activity

5 × 10^5^ DC were incubated at 37 °C for 2 h with FITC labeled dextran (MW 40,000 Da, Sigma Aldrich, 3 mg/mL). Afterwards, cells were extensively washed with PBS and analyzed by flow cytometry. Extracellular FITC signal was assessed by incubating cells on ice during incubation with FITC dextran.

### 2.6. Allogeneic Mixed Lymphocyte Reaction (MLR)

Peripheral blood mononuclear cells (PBMCs) were isolated from donor blood by Ficoll-Paque density gradient centrifugation and resuspended in CellGRO medium. Unidirectional MLRs were performed by coculturing 1 × 10^5^ PBMCs (responder cells) with stimulating cells (S) in a 96-well plate (Corning, New York, NY, USA). S cells were represented by 1 × 10^4^ DCs, 1 × 10^5^ autologous PBMCs (for auto-MLR, negative control), or 1 × 10^5^ allogeneic PBMCs (for allo-MLR, positive control). S cells were pretreated with mitomycin-C (50 mg/mL; Sigma-Aldrich) for 20 min at 37 °C and used after extensive wash. The colorimetric nonradioactive MTT assay (Promega Corporation, Madison, WI, USA) was used for proliferation detection. Briefly, after 5 days of culture, the MTT reagent was added as one-tenth of the volume per wells for another 4 h. Cell proliferation was assessed by evaluating absorbance at wavelength of 540 nm.

### 2.7. Migration Assay

Mature DC (2 × 10^4^) were seeded in a 4 μm insert (Corning) in 200 μL medium; in the lower chamber 600 μL medium containing 100 ng/mL of CCL21 (Sigma Aldrich) were added. After 3 h incubation at 37 °C, cells migrated into the lower compartment were collected. Samples were analyzed on a FACS Calibur (BD) after addition of propidium iodide (Sigma-Aldrich) in order to discriminate dead cells. Counting beads (Beckman Coulter, Brea, CA, USA) were added to the samples directly before measurement in order to normalize acquisition rates among the samples.

### 2.8. Proliferation Assay

Briefly, autologous PBMCs were isolated by FicollPaque gradient (GE Healthcare). Autologous immature DC (iDC) were incubated with or without Hemocyanin, Keyhole Limpet from Megathura crenulata (KLH, 5 µg/mL, Calbiochem) for 24 h before they were induced to maturation. For proliferation tests the DC:T cell ratio was 1:10 for in vitro stimulation of 5 days prior to MTT analysis. The culture conditions were 37 °C, 5% CO_2_. Wells containing T cells alone (no stimulus) or with non-pulsed DC were included as negative controls.

### 2.9. Statistical Analysis

Results are expressed as mean ± SD. One-way analysis of variance and a two-tailed test were used for all statistical analyses and performed with GraphPad Prism software version 4.0 (GraphPad Software Inc., San Diego, CA, USA). *P* values of less than 0.5 were considered significant.

## 3. Results

### 3.1. Yield, Morphology, and Phenotypic Characteristics

The feasibility of generating Fast-DC pulsed with whole tumor lysate was assessed using a series of small-scale cultures performed in parallel with clinical grade large scale standard method preparations (*n* = 8). Cell culture media, cytokines and closed-system containers were selected for direct translation of Fast protocol to GMP production. We evaluated the role of PGE_2_ for the success of the final product.

The Fast protocol resulted in the generation of a population showing the characteristic dendritic cells morphology as evaluated using light microscopy, although Fast-DC were considerably smaller and less granular than standard DC. At harvest, mDC-F resulted strongly adherent to culture surface respect to mDCp-F (Figure 1).

DC yield and viability, pre and post cryopreservation, were also included in the evaluation of the Fast method, as reported in Table 1. Fast-DC resulted in a considerably higer yield respect to standard method both in presence or in absence of PGE_2_ (mDC 8.26 ± 2.67 vs. mDC-F 20.07 ± 7.90, *p* < 0.05; mDCp 9.30 ± 3.43 vs. mDCp-F 25.20 ± 7.60, *p* < 0.05).

Phenotypic characterization of Fast versus standard DC was made by FACS analysis. Several surface markers including maturation markers and co-stimulatory molecules for T cell activation have been considered. Fast-DC showed down-regulation of CD14 and high levels of MHC II expression (>95% expression for all DC types) consistent with early immature DC development; moreover cells displayed a fully mature DC immune-phenotype (CD83, CD80, and CD86 high). Although a moderate increase in the forward/side scatter intensity of Fast-DC compared with monocytes could be observed, they were considerably smaller and less granular than standard DC (data not shown).

Notably Fast method performed without PGE_2_ fail in inducing upregulation of CD80, CD83, and CD86 maturation marker. FACS data are reported in Figure 2.

### 3.2. Comparison of Endocytic Activity

The ability to take up and process antigen is a hallmark of immature DC functions and is rapidly lost upon maturation of DC [11]. To assess the uptake of soluble antigens via endocytosis, unloaded Fast-DC and standard method DC were incubated with FITC-conjugated dextran at 37 °C for 2 h. Dextran uptake was determined by FACS analysis. Fast immature DC take up dextran (Figure 3) with an efficiently comparable to standard immature DC as reflected by MFI (59.73 ± 9.30 iDC-F; 61.57 ± 11.14 iDC, *p* = ns); activation with proinflammatory cytokines lead to a rapid reduction of dextran uptake as a result of DC maturation (mDCp-F 10.73 ± 0.32, mDCp 14.28 ± 2.11).

### 3.3. Migration Assay

Functional analysis of the Fast versus standard DC was assessed by a migration assay toward CCL21. Both Fast and standard DC possess high migratory capacity and were capable of specifically migrate under the chemokine gradient (Figure 4). Fast-DC cultivated without PGE_2_ results in a lower migration activity (number of migrating cells ± SD: mDCp-F 6289 ± 989; mDC-F 2600 ± 400; *p* < 0.05).

### 3.4. IL12 Secretion

The secretion of IL-12 (p70) was tested using supernatants from the day of harvest. The concentration of IL-12 (p70) measured in Fast as well as standard DC was, as expected, low due to the presence of PGE_2_ in the maturation cocktail [20] (mean pg/mL ± SD mDCp-F 0.68 ± 0.54; mDCp 0.59 ± 0.49). To test our DC preparations for optimal cytokine secretion capacity LPS was used for stimulation; however, this resulted in a very few increase in secretion of IL-12p70 compared to maturation cocktail alone (data not shown).

### 3.5. PBMC Stimulatory Capacity

The T cell stimulatory capacity of Fast-DC was assessed and compared with that of standard DC using KLH as a model Ag. For proliferation tests the DC:T cell ratio was 1:10 for in vitro stimulation of 5 days prior to MTT analysis. Ag-loaded mDCp-F were effective in inducing the proliferation of autologous PBMC isolated from DC-vaccinated patients with a comparable potency respect to Ag-loaded mDCp (Figure 5). Unloaded DC did not induce proliferation.

The capacity to promote T cell proliferation was also assessed in an allogeneic MLR, as MLR is the gold standard to test the functional ability of DCs as antigen-presenting cells [21] outside the context of antigen-specificity. T cells were co-cultured with any of the 4 types of DC (mDC, mDCp, mDC-F and mDCp-F) for 5 days and proliferation was measured by MMT test. The results in Figure 6 show that the 4 types of DC are equally powerful in inducing primary allogeneic MLRs.

## 4. Discussion

DC is an attractive target for therapeutic manipulation of the immune system to enhance insufficient immune responses in aggressive cancer such as glioblastoma.

Thus far, numerous phase I/II clinical trials with DC vaccines have been performed in cancer patients [7,22,23,24], concluding that DC vaccination strategies are safe, well tolerated, and capable of inducing tumor antigen-specific immune responses. Despite promising successes, a lot of challenges remain.

Differentiation of DC from monocytes obtained from leukapheresis is by far the most commonly used approach to generate DC-based medicinal products suitable for clinical applications. CD14+ monocytes are usually positive selected using immunomagnetic beads, then differentiated through controlled supplementation of cytokines; obtained immature DC are loaded with tumor antigens and finally stimulated to mature by exposure to inflammation cocktail.

Dauer and colleagues on 2003 described a strategy for differentiation and maturation of monocyte derived DC within only 48–72 h of in vitro culture, called “Fast” protocol, and several other groups have further elaborated on this pioneering work [13,14,15,16,17]. Here we described this method modified from our group to obtain Fast mature autologous DC, loaded with antigens obtained from whole tumor lysate, suitable for the immunotherapy of glioblastoma patients. A Table that recapitulate the previously published Fast protocols characteristics respect to our new Fast method is reported in the Appendix A. The validation of our new Fast protocol is of interest in the contest of DC-based clinical trials because it takes into consideration both the gold standard maturation cocktail (the most used in therapeutic approaches) and the DC’s antigen-loading phase which is often “omitted” in most of the previously published protocols. Furthermore, the absence of sera in the culture medium used in our protocol facilitates GMP translation.

Tumor cell lysate represents the whole protein content of lysed tumor cells. Utilizing whole tumor cells as a source of antigen for DC it ensure that the entire repertoire of antigens associated with a given tumor can be processed. This is useful especially in those tumor that do not have molecularly defined antigens and could prevent tumor immune escape. Our method to generate tumor lysates is designed to conserve both lipid-soluble and water-soluble molecules as tumor lysis takes place in 0.9% NaCl without detergents [19].

Our innovative short protocol has been compared with the standard protocol developed in our laboratory and used for the production of DC for the treatment of glioblastoma patients in the contest of a phase I clinical trial, starting from 2009 and steel recruiting (Eudract n. 2008-005035-15).

Production of DC for therapeutic purposes has to fulfill several aspects regarding DC function such as up-regulation of maturation markers, expression of co-stimulatory molecules, capacity to prime T cells responses and ability to home to lymph nodes.

Regarding the maturation stimuli, Fast method was compared to standard culture, both in presence or in absence of PGE_2_ due to our variation to the clinical gold standard cytokine cocktail that, since 2016, omitted the use of PGE_2_.

The results presented here demonstrate that our Fast-DC protocol generate DC of comparable quality to standard method.

Concerning the phenotype, as determined by the expression of relevant surface markers, Fast-DC maturated in the presence of PGE_2_ were almost comparable to the standard DC (with or without PGE_2_). Conversely, Fast-DC maturated without PGE_2_ fail in the upregulation of maturation markers costimulatory molecules.

Fast-DC loaded with KLH and matured in the presence of PGE_2_ induced equal rates of autologous T cell proliferation compared with DC generated according to a 7-day standard protocol. KLH is a strongly immunogenic protein used as a helper antigen. A concomitant presentation of KLH and tumor antigens by the same DC strongly augmented the weak tumor reactive T lymphocytes proliferation [25,26]. For this reason, the use of KLH during DC-antigen loading may improve the efficacy of therapeutic anti-tumor vaccination. KLH also serves as a tracer molecule to monitor the immune response. Positive in vitro T cells responses obtained from autologous PBMC from vaccinated patents indicate, first of all, that our standard vaccination protocol is efficient in the induction of antigen-specific immune response in GBM patients. Moreover our Fast-DC resulted qualitatively identical to the standard one in stimulating T cell, in vitro, demonstrating that Fast protocol generate DC that are capable of taking up and processing tumor proteins for presentation on MHC molecules. The ability to process soluble antigens was also confirmed as DC (both Fast and standard) efficiently endocytosed dextran in their immature state.

The capacity to promote T cell proliferation was also assessed in allogeneic MLRs: results show similar T cell stimulatory capacity between the DC measured, confirming that Fast-DC acquired T cell stimulatory capacity comparable to standard DC.

Fast-DC cultivated in presence of PGE_2_ are sensitive to chemokines CCL21, possess a high migratory capacity comparable to standard DC, thus demonstrating all the qualities currently thought to be necessary for an effective DC-based antitumor vaccine. The absence of PGE_2_ results in the impossibility to migrate for Fast-DC, confirming that PGE_2_ is essential for the adequate maturation of DC in the Fast protocol. On the contrary standard 7-days-long protocol mDC results able to migrate despite the absence of PGE_2_ in the maturation cocktail. This can be explained by Muthuswamy and colleagues [27] that demonstrate that CCR7 expression induced by PGE_2_ exposure on standard DC were rapidly compensated after the harvesting of DC from the maturation cultures and replating them in neutral conditions in the presence of GM-CSF only: this compensation of the differences in surface CCR7 expression was reflected by the elimination of differences in migratory capacity in vitro between PGE_2_-matured DC and DC matured in absence of PGE_2_.

DC stimulated with pro inflammatory mediators (TNF-α, IL-1β, IL-6, and PGE_2_) did not produce (or produce very low levels of) IL-12p70. Failure of Fast-DC activated with proinflammatory cytokines plus PGE_2_ to secrete IL-12(p70) is not surprising if some authors previously reports on the effect of PGE_2_ on cytokine production. It has repeatedly been shown that PGE_2_ inhibits the secretion of IL-12 (p70) by DC even if potent stimuli, such as CD40L, are used for activation [28]. IL-12 DC’s production is (and will be in the future) a controversial theme. Despite the inhibitory effect on IL-12, it has been demonstrated that PGE_2_ per se is important for DC’s ability to stimulate T cells; moreover it induce high expression of CCR7 resulting in strong migration ability [29,30]. Several vaccination studies have obtained induction of T cell responses despite the use of PGE_2_ on injected DC [31,32]; on the contrary we also demonstrated that DC matured in the absence of PGE_2_ are able to induce Ag specific T cell responses in treated patients. So PGE_2_ may be considered not fundamental for the production of standard 7-day-long efficient mature DC for immunotherapy. On the other hand the presence of PGE_2_ results mandatory for the success of Fast method in which the “standard” maturation cocktail (TNF-α, IL-1β, IL-6, and PGE_2_) can be considered a good “compromise” between IL-12 production and DC complete maturation.

Our findings suggest that the Fast protocol used for the production of DC loaded with whole tumor-lysate derived antigens is feasible. The functional characterization showed that these protocol is effective only in the presence of PGE_2_ in the maturation cocktail to guarantee that Fast-DC cells exhibit a mature phenotype and fulfill all requirements for in vivo use in immunotherapy approaches. Moreover, our data show that a Fast-DC loaded with whole tumor-lysate antigens stimulate, in vitro, lymphocytes isolated from vaccinated patients with a potency comparable to standard DC.

Keeping it simple our Fast protocol could be successfully translated to a GMP-compatible technology in a running vaccination program. This Fast protocol reduces the time, workload, and cost associated to GMP grade production and thus may facilitate the use of DC in clinical trials, opening the possibility of larger, multicenter immunotherapy trials involving other facilities.

## Figures and Tables

**Figure 1 pharmaceutics-12-00215-f001:**
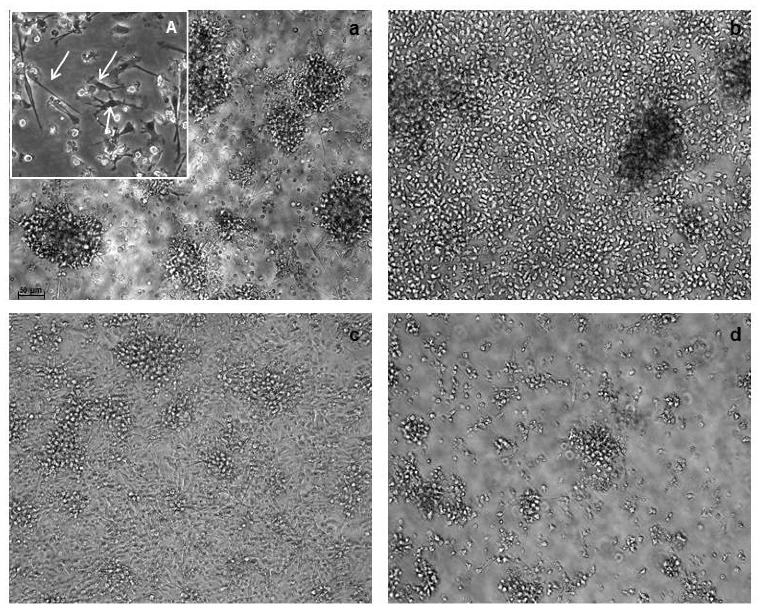
Morphological characterization of dendritic cells (DC). Images are from light microscopy at 20× magnification. Fast-DC cultivated in absence of PGE_2_ (**a**) resulted strongly adherent to culture surface respect to Fast-DC cultivated with PGE_2_ (**b**). Adherent cells are evident and point out by arrows in 40× magnification image (A). Pictures (**c**,**d**) represent standard method DC respectively without PGE_2_ and with PGE_2_. The absence of PGE_2_ results in higher percentage of adherent cells also in the standard method.

**Figure 2 pharmaceutics-12-00215-f002:**
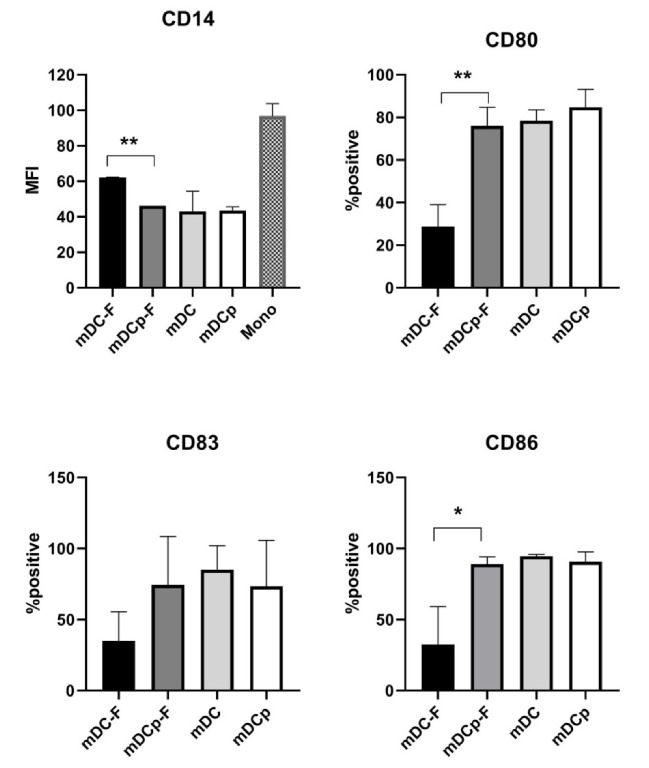
Phenotypic analysis of DC. Summarizing graphs represent the results of the FACS analysis on Fast and standard method DC both in presence (mDCp-F, mDCp) or absence (mDC-F, mDC) of PGE_2_. Down-regulation of CD14 (monocytes marker) is represented as MFI; variation of maturation markers (CD80, CD83, CD86) is represented as mean percentage of positive cells. Statistical bars on graph indicate the SD value. Significance denoted, * *p* < 0.05, ** *p* < 0.01.

**Figure 3 pharmaceutics-12-00215-f003:**
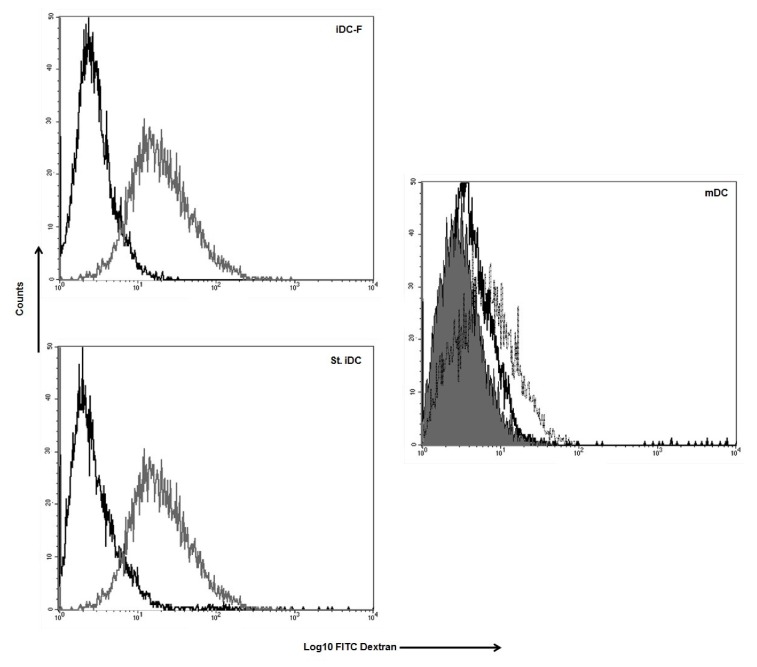
Uptake of FITC labeled dextran by DC. Histograms show the FACS analysis of the cell co-cultured with FITC labeled dextran at 37 °C (grey line) and non-specific labeling of the cells (black line) for iDC-F (immature Fast-DC); St. iDC (standard method immature DC). Mature DC (mDC) were analysed as negative control for dextran uptake (dotted line Fast method; black line standard method; solid line isotype). One representative performed experiment is presented.

**Figure 4 pharmaceutics-12-00215-f004:**
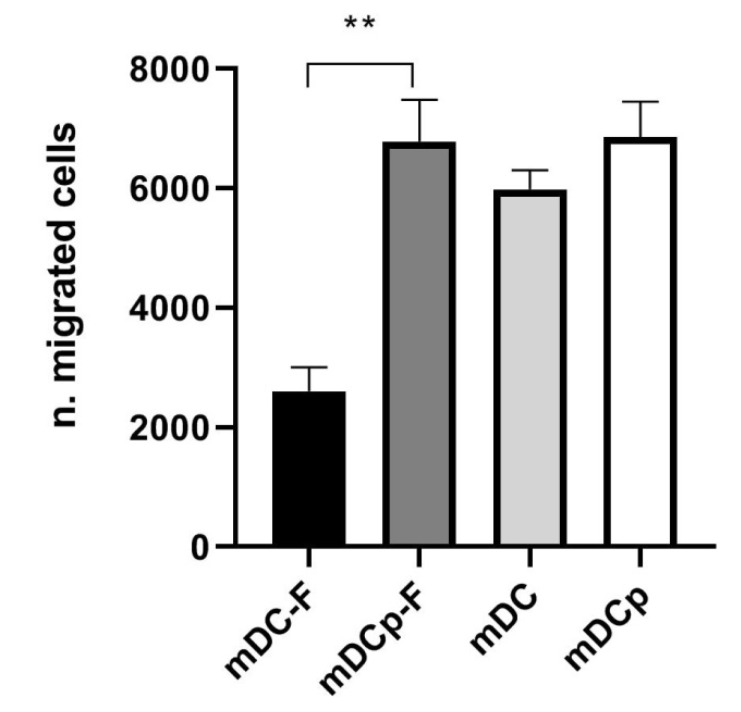
CCL21 induced migration. The ability of mature DC to migrate toward the CCR7 ligands CCL21 was evaluated in a trans-well migration assay. After 3 h incubation at 37 °C, DC migrated in the lower chamber were counted by flow cytometry. The results of three experiments are represented as mean ± SD. ** *p* < 0.01.

**Figure 5 pharmaceutics-12-00215-f005:**
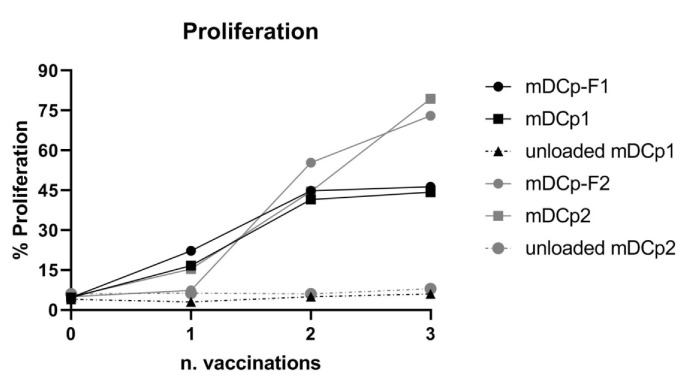
Antigen specific proliferation. Keyhole limpet hemocyanin (KLH)-loaded Fast-DC and KLH-loaded standard method DC induce Ag-specific proliferation of autologous peripheral blood mononuclear cells (PBMCs) isolated from DC-vaccinated patients. Unloaded Fast and standard DC were used as control for aspecific proliferation. The graph represent the proliferation of PBMCs, from two different patients, obtained during 3 sequential DC administrations. Baseline is represented by proliferation measured pre-vaccination (n = 0).

**Figure 6 pharmaceutics-12-00215-f006:**
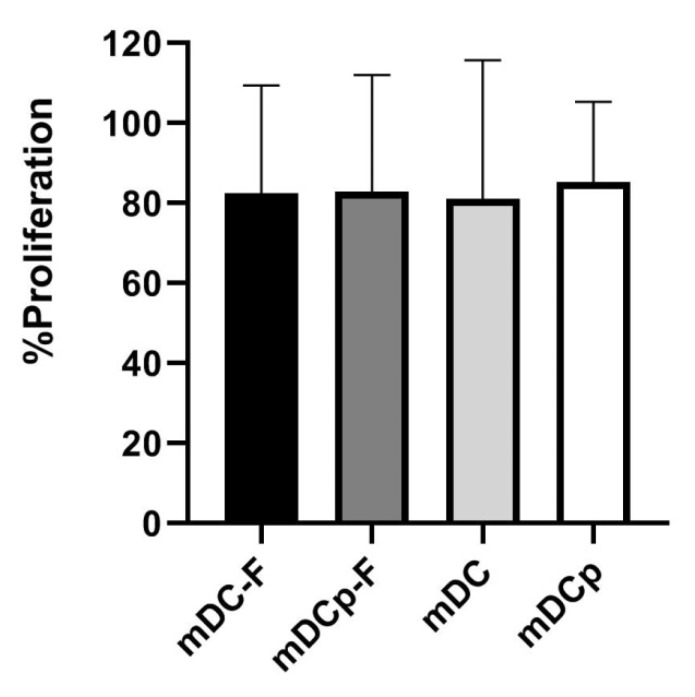
Mixed lymphocytes reaction. DC functionality was evaluated via one-way allogeneic mixed lymphocyte reaction (MLR). Colorimetric variation of MTT reagent was used to detect cell proliferation. All DC analyzed were potent stimulatory cells; the comparison between Fast and standard method showed that there are no differences in DC’s ability to induce allogeneic MLR. Bar graphs show the mean proliferation ± SD.

**Table 1 pharmaceutics-12-00215-t001:** Comparison of total cell yield, cell viability, recovery and viability after thawing of DC from Fast and standard methods.

Cell Type	DC/CD14 (%)	Viability (%)	Recovery after Thawing (%)	Viability after Thawing (%)
mDC	8.26 ± 2.67	82.7 ± 7.1	98.3 ± 4.5	90.0 ± 6.0
mDCp	9.30 ± 3.43	88.3 ± 9.3	93.3 ± 4.6	92.7 ± 3.8
mDC-F	20.07 ± 7.90	94.0 ± 4.0	95.5 ± 4.7	93.3 ± 4.2
mDCp-F	25.20 ± 7.60	96.3 ± 1.2	83.6 ± 12.2	89.3 ± 12.5

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
