# Peer review of "PGE_2_ Is Crucial for the Generation of FAST Whole- Tumor-Antigens Loaded Dendritic Cells Suitable for Immunotherapy in Glioblastoma"

_pharmaceutics, 2020, doi:10.3390/pharmaceutics12030215_

Round 1

Reviewer 1 Report

This study is of high significance, well-designed and presented.

In the Introduction, the authors mention "several groups have shown that it is possible to obtain mature DC in 2–3 days cell culture [13-17]". In their study, the authors present one more fast DC method. However, it is not clear from the manuscript why the scientific community should to have one more fast method to obtain mature DC. The authors should compare in detail their method with previously published fast DC protocols and explain possible advantages and drawbaks. This comparison could be presented as an additional Figure/table/scheme.

Author Response

We thank the Reviewer for the generous comments. We have edited the manuscript to address his concern. The Reviewer can find the modification on the discussion paragraph (yellow underlined). Please see the attachment for Table S1 and revised manuscript.

Best regards

Sara Nava

Reviewer 2 Report

Nava Et al. reported a fast protocol for the production of DC loaded with whole tumor lysate, which could be potentially translated to a GMP-compatible platform. The manuscript is well written, the topic was clearly explained and put into context. Experimental strategy and results were clearly described and well presented in a logical order.

Author Response

We thank the Reviewer for the generous comments.

Please see the attachment for revised manuscript. The Reviewer can find the modification on the discussion paragraph (yellow underlined).

Best regards

Sara Nava
